# Streamflow Variability Indicated by False Rings in Bald Cypress (*Taxodium distichum* (L.) Rich.)

**Matthew D. Therrell** [1,*], **Emily A. Elliott** [1], **Matthew D. Meko** [2], **Joshua C. Bregy** [3,4],
**Clay S. Tucker** [1], **Grant L. Harley** [5], **Justin T. Maxwell** [3] **and Glenn A. Tootle** [6]

1. Department of Geography, University of Alabama, Tuscaloosa, AL 35487, USA; emily.elliott@ua.edu (E.A.E.); cstucker3@ua.edu (C.S.T.)
2. Department of Geosciences, University of Arizona, Tucson, AZ 85721, USA; meko@email.arizona.edu
3. Department of Geography, Indiana University, Bloomington, IN 47405, USA; jbregy@indiana.edu (J.C.B.); maxweljt@indiana.edu (J.T.M.)
4. Department of Earth and Atmospheric Sciences, Indiana University, Bloomington, IN 47405, USA
5. Department of Geography and Geological Sciences, University of Idaho, Moscow, ID 83844, USA; gharley@uidaho.edu
6. Department of Civil, Construction and Environmental Engineering, University of Alabama, Tuscaloosa, AL 35487, USA; gatootle@eng.ua.edu
* Correspondence: therrell@ua.edu

**Abstract:** Despite growing in wet lowland and riparian settings, *Taxodium distichum* (L.) Rich. (bald cypress) has a strong response to hydroclimate variability, and tree ring chronologies derived from bald cypress have been used extensively to reconstruct drought, precipitation and streamflow. Previous studies have also demonstrated that false rings in bald cypress appear to be the result of variations in water availability during the growing season. In this study 28 trees from two sites located adjacent to the Choctawhatchee River in Northwestern Florida, USA were used to develop a false ring record extending from 1881 to 2014. Twenty false ring events were recorded during the available instrumental era (1931–2014). This record was compared with daily and monthly streamflow data from a nearby gage. All 20 of the false-ring events recorded during the instrumental period occurred during years in which greatly increased streamflow occurred late in the growing season. Many of these wet events appear to be the result of rainfall resulting from landfalling tropical cyclones. We also found that the intra-annual position of false rings within growth rings reflects streamflow variability and combining the false-ring record with tree ring width chronologies improves the estimation of overall summer streamflow by 14%. Future work using these and other quantitative approaches for the identification and measurement of false ring variables in tree rings may improve tree-ring reconstructions of streamflow and potentially the record of tropical cyclone rainfall events.

**Keywords:** *Taxodium distichum*; bald cypress; dendrochronology; tree rings; intra-annual density fluctuation

## 1. Introduction

False rings (FRs) or inter-annual density fluctuations are generally identified by a thin band of radially narrower, thick-walled tracheids within a wider annual band of earlywood type tracheids or earlywood-like cells in the latewood [1,2] The occurrence of FR in various tree species has been well documented [3–16], and although highly variable depending on the growth conditions favorable to each species, the formation of FR has largely been shown to frequently result from rapid reversals in environmental conditions, such as drought followed by increased rainfall [2,10,14]. While these bands can often be mistaken for annual or latewood ring boundaries, therefore complicating

dendrochronological analysis [11], it has also been shown that these anatomical anomalies can be a useful proxy for analyzing specific hydroclimatic conditions [6–10,15–18].

Although several studies have utilized the strong hydroclimate response of bald cypress (*Taxodium distichum* (L.) Rich.) to reconstruct a variety of climate variables and large-scale climate forcing mechanisms [13,19–24], and the annual growth rings of southern bald cypress frequently contain FR [3–5,16], comparatively little research has been focused on the climate or meteorological signal embedded in the FR of bald cypress. We are aware of only two studies that have investigated the relationship between environmental variability and FR formation in bald cypress and the potential to use these signals to better understand hydroclimate variability [5,16].

Young et al. [5] investigated the formation of FR in juvenile bald cypress resulting from prolonged inundation and found that FR occurrence is common in bald cypress saplings, resulting from intermittent inundation stress, but in mature trees, radial growth has been shown to be strongly influenced by streamflow variability rather than inundation stress [25,26]. Copenheaver et al. [16] further demonstrated that in mature bald cypress, FR appear to be associated with changes in streamflow, especially in late summer, when tree ring width is often a weak predictor of hydroclimatic variability [22]. Identifying the specific relationship between FR and hydroclimatic variability should provide a better understanding of the species' response to short-term variability as well as potentially allowing the use of these anatomical anomalies to improve reconstructions of climate and therefore streamflow [8].

In this study, we compare a record of FR derived from bald cypress trees growing along the Choctawhatchee River in Northwestern Florida, USA to instrumental records of daily and monthly streamflow and potentially associated tropical cyclone events. We also use the FR record in conjunction with records of tree ring width to estimate past seasonal streamflow in order to determine whether this approach may improve estimates based on tree ring width alone. Given the importance of bald cypress as a paleoenvironmental proxy, improved understanding of this species' subseasonal environmental response may provide important new ecological and paleoclimatic information relevant to the study of forest and stream ecology, tree ecophysiology and climate variability.

## 2. Materials and Methods

### 2.1. Study Area

The Choctawhatchee River basin spans 11,000 km$^2$ in Southeastern Alabama and Northwestern Florida (Figure 1).

It is one of the longest unimpaired rivers in the USA (230 km), and the average annual discharge is approximately 197 m$^3$/s at the Bruce, Florida gage [27]. We collected samples at two sites (John's Lake (JLK) and Bruner's Island (BRI)) near the terminus of the river into the Choctawhatchee Bay estuary, (30.442154° N, 85.907353° W; Figure 1). This area was previously sampled by Stahle and Cleaveland [28], allowing collected samples to be cross-dated against this established record. The JLK site is an oxbow lake adjacent to the river, and the BRI site includes samples taken along the shore of an island surrounded by the river approximately 1 km from the JLK site (Figure 1).

The climate of this region is humid subtropical, with hot summers (29 °C average for July) and mild winters (9 °C average for January). Precipitation in the area averages around 1700 mm and occurs largely as a result of convective thunderstorms in summer and landfalling tropical cyclones in summer and fall [29]. Soils at these sites are frequently flooded, poorly drained Typic Fluvaquent Entisols of the riparian floodplain swamp [30] at an elevation of approximately 3 m above msl. Monthly climate and hydrography are shown in Figure A1.

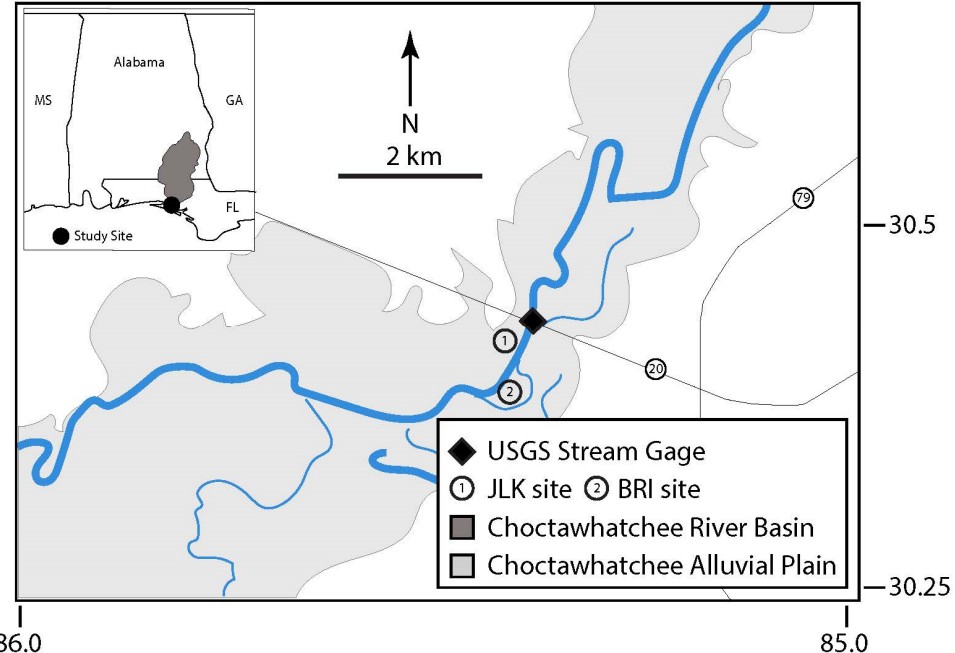

**Figure 1.** The Choctawhatchee River basin is shown in the inset panel (**dark grey**). The river channel (**blue line**) and alluvial plain (**light grey**) are shown in the primary panel along with the tree ring collection site at John's Lake and Bruner's Island (**numbered circles**); and the United States Geological Survey (USGS) gage near Bruce, Florida (**black diamond**).

*2.2. Tree Ring Data*

We collected 1–4 increment core samples from 46 (JLK) and 10 (BRI) living bald cypress trees in January 2015. Increment core samples were collected using a 5.15 mm Swedish increment borer and were taken above the basal swell at approximately 2 m height. All increment cores were air dried, mounted and sanded until cellular structure was visible under a microscope [31]. We used the skeleton plot method of cross-dating patterns of relative ring width to determine the exact year of annual rings [32]. We measured earlywood (EW), latewood (LW) and total ring width (TRW) to the nearest 0.001 mm on a stage micrometer using the criteria for determination of the boundary between earlywood and latewood described by Stahle et al. [33,34]. We then used the computer program COFECHA to statistically verify accurate sample dating and measuring [35]. We also used COFECHA to compare our ring width data to the Stahle [28] collection to verify our cross-dating. We then computed the standardized (mean-value index) chronologies from the EW, LW and TRW measurement series using the package dplR in the R statistical computing environment [36,37]. We also computed a suite of common chronology statistics including the mean correlation within trees ($\bar{r}_{wt}$), mean correlation between trees ($\bar{r}_{bt}$), effective mean correlation ($\bar{r}_{eff}$) and expressed population signal (EPS; [38]). Finally, we computed the "adjusted LW" ($LW_{adj}$) chronology to remove any intercorrelation between EW and LW [33,34,39]. We then examined each annual ring for evidence of FR occurrence. The criteria we used to identify FR was simply the visual appearance of a transition from EW to latewood-like cells and a return to EW-like cells (Figure 2). In addition to recording the presence of a FR we also recorded the relative position of the FR within the annual ring. Specifically, we described the position as "early" if the FR occurred in the first 1/3 of the ring, "middle" if it occurred in the middle 1/3 and "late" if in the final 1/3 (e.g., [14,40]). In years with multiple FRs, we based our categorization on the first appearance of a FR in the annual ring (e.g., years with early and middle FR would be categorized as early). Based on this analysis we calculated the proportion (F) of all trees exhibiting a FR (N) compared to all 28 trees analyzed (n) for each year (1931–2014) as shown in Equation (1) [8,41].

$$F = N/n, \tag{1}$$

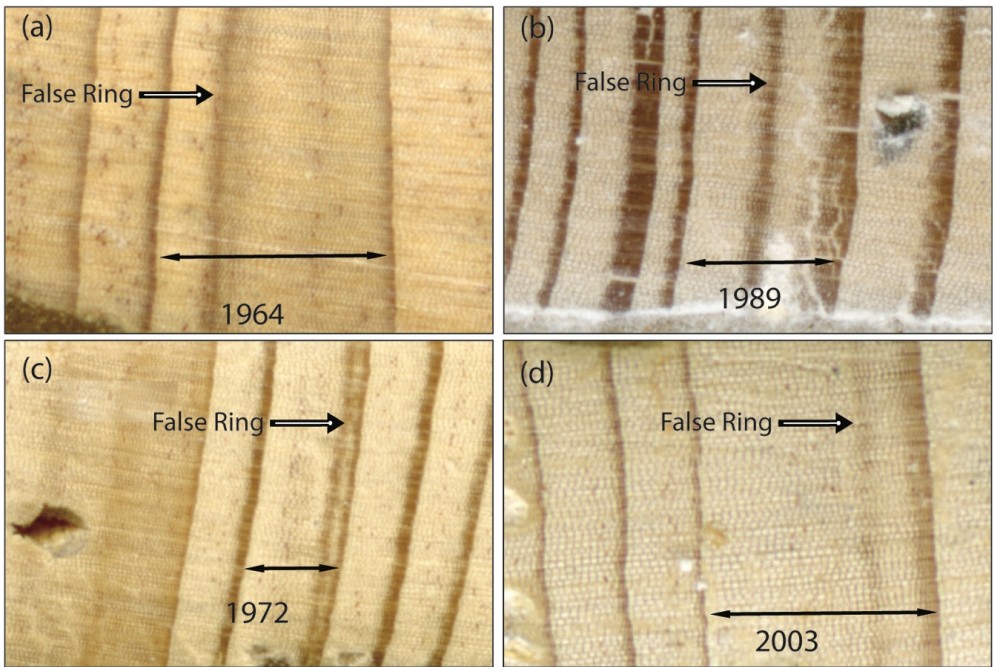

**Figure 2.** Microphotographs of *Taxodium distichum* annual rings (highlighted) showing false ring anatomy during four separate years. An "early" false ring (FR) is shown in (**a**), a "middle" FR in (**b**) and "late" in (**c,d**).

We categorized years in which the proportional values were in the top quartile of all years (≥20% of trees produced a FR) as "high-FR" (HFR) years.

To use the FR data in a generalized linear regression model to estimate monthly streamflow data, we assigned a binary value of 1 to FR years that fell within the top quartile of all F values, and 0 to those that fell below this threshold. As discussed in Wimmer [8], the benefit of using a 'binary variable' is that it can be treated statistically as an interval level variable, allowing for a 'dummy' variable to be used in regression equations, which can be used as a proxy for the presence or absence of a FR in a given year.

## 2.3. Climate and Streamflow Data

We compared our EW, LW, $LW_{adj}$ and TRW chronologies and our HFR index chronology to daily and monthly Choctawhatchee River streamflow (Q) using data obtained from the United States Geological Survey (USGS) gage near Bruce, FL (Gage # 02366500; 30.450833° N, 85.898333° W; Figure 1; [27]). This gage is located about 1 km upstream from the sample sites and provides complete daily and monthly discharge over the period 1931–2019, except for limited missing data in 1983 and 1984, which were replaced with values extrapolated from a linear model of flow between the gages near Bruce and Caryville, FL (Gage # 02361000; 30.70888° N, 85.82777° W).

To evaluate the potential association between monthly discharge and the 20 HFR years in the instrumental period (Table 2), we used Rao et al.'s [42] double bootstrap approach to carry out a superposed epoch analysis (SEA) for each month from March through to September over the 1931–2014 period. We used 1000 unique draws of 15 of the 20 HFR years compared to 10,000 draws of 15 "pseudo HFR years" chosen at random.

Evaluation of the relationship between HFR and daily streamflow data was limited to visual comparisons of time series of daily streamflow in individual HFR years, average daily streamflow in HFR years and non-HFR years, years of both wide (>) and narrow (<; chronology average) ring width associated with HFR years, and daily streamflow plotted according to placement of FR within the annual ring.

### 2.4. Regression Equations: Using Ring Width and False Rings to Predict Summer Streamflow

Using a generalized linear model (GLM), we modeled streamflow over the instrumental period (1931–2014) using TRW and a binary FR variable that distinguishes the presence or absence of a FR in a given year [8]. Using the GLM, streamflow was estimated using the following linear equation:

$$y = b_0 + b_1 X_1 + b_{2\times2}, \tag{2}$$

in which y is the predicted or expected value of the dependent variable, $X_1$ and $X_2$ are the distinct independent or predictor variables, $b_0$ is the intercept, and $b_1$ and $b_2$ are the estimated regression coefficients (slope) of $X_1$ and $X_2$, respectively. In this case, y is June, July or August total discharge or total discharge for those three months (JJA), $b_0$ is the coefficient for the intercept, $X_1$ and $X_2$ represent TRW and the binary false rings variable ($FR_b$), and $b_1$ and $b_2$ are the coefficients for the slope of TRW and $FR_b$, respectively. The $FR_b$ coefficient ($b_2$) is the estimate of the difference in groups with and without false rings, and therefore indicates the change in JJA discharge/streamflow when FRs are present in a given year. Since streamflow values showed significant outliers, all values of JJA were log-transformed prior to statistical analysis, and subsequently followed a normal distribution. GLMs for the log-transformed JJA streamflow are compared using TRW alone, relative to the model where both TRW and the $FR_b$ are included and are presented in Table 3, with inverse-transformed modeled values presented for each modeled discharge (TRW and TRW + $FR_b$) against observed total JJA discharge in Figure 8.

### 2.5. False Ring Association with Summer Storms/Tropical Cyclones

Research has suggested that FR in bald cypress may indicate tropical cyclone (TC) activity in the Southeastern U.S. during the mid to late summer months [16]. To explore this idea, we examined all May–September TC activity within 50, 100 and 200 nautical miles (NM) of the center of the Choctawhatchee watershed (Figure 1) using the NOAA Historical Hurricane Tracks Tool, which is based on International Best Track Archive for Climate Stewardship (IBTrACS [43,44]), and the National Hurricane Center (HURDAT2; [45]) and compared these events to the 20 HFR years. We also compared HFR years to the NOAA Tropical Cyclone Rainfall dataset ("Roth" [46]) and because the Roth dataset does not extend back to 1931 we also used a similar catalog of tropical rainfall compiled by Schoner and Molansky [47] that covers that period.

## 3. Results

### 3.1. Ring Width Chronology

We were able to correctly cross-date 51 series (28 trees) from the two sites. The chronology statistics for both TRW and EW chronologies were similar and quite strong (Table 1). Statistics for the LW chronology were generally much weaker (Table 1), but given that the chronologies were accurately dated and we wished to examine the appropriately comparative datasets, we made no effort to improve the LW chronology by removing series (or portions thereof) to improve the chronology.

**Table 1.** Chronology statistics for total ring width (TRW) and earlywood (EW) and latewood (LW) indices [1].

|     | $n_t$ | $n_{\bar{t}}$ | $c_{eff}$ | $\bar{r}_{wt}$ | $\bar{r}_{bt}$ | $\bar{r}_{eff}$ | EPS |
|-----|-------|-------|-------|-------|-------|-------|-----|
| TRW | 28 | 14.6 | 1.68 | 0.68 | 0.41 | 0.47 | 0.93 |
| EW  | 28 | 14.6 | 1.69 | 0.68 | 0.41 | 0.47 | 0.93 |
| LW  | 28 | 14.6 | 1.69 | 0.35 | 0.14 | 0.19 | 0.77 |

[1] Total number of trees ($n_t$), average number of trees contributing to each year of the chronology ($n_{\bar{t}}$), effective number of cores per tree ($c_{eff}$), mean correlation within trees ($\bar{r}_{wt}$), mean correlation between trees ($\bar{r}_{bt}$), effective mean correlation ($\bar{r}_{eff}$) and effective population signal (EPS) of 51 standardized series each of TRW, EW and LW measurements contributing to the mean index chronologies.

## 3.2. False Ring Chronology

The full record length of the 28 cross-dated trees analyzed extended from 1801 to 2013 and within that period 372 of the 5099 annual rings included a FR (i.e., a proportion of 7%), which indicates the expected random background occurrence of a FR [10,48]. Of the 372 FR rings, only 7% of FR were classified as "early", 24% were "middle" and 69% were "late". Additionally, at least one FR was present in all the cores that were analyzed. We confined our present analysis to the 83-year period of overlap between the available instrumental streamflow data for the site and our FR chronology (1931–2014; Table 2). During that time there were 20 years in the top quartile of FR years, which we used to define "high" FR (HFR) years (e.g., [8]). These events are roughly evenly distributed through time with about one-to-three events per decade on average (Figure 3). The position of the FR within rings was classified as "middle" for 7 of the HFR years, while the remaining 13 HFR years were classified as "late". While these positions varied to some extent among and even within trees, values were assigned based on the modal position for all trees in each FR year (Table 2). Comparison of instrumental period HFR years with total TRW indicates that well over half (14 of 20) HFR years were above average TRW and EW. Sixteen of the 20 years were above average LW, but only 10 were wide LW$_{adj}$ (Table 2).

**Table 2.** Ranked list of 20 years (top quartile of all years) with the highest proportion of trees indicating a false ring and FR location and associated ring width values (mm) and tropical cyclone events [1].

| Year | FR% | Type | EW | LW | TRW | TC 50 NM | TC Rainfall Data |
|------|-----|------|-----|-----|-----|----------|------------------|
| 1989 | 65 | M | 1.707 | 0.917 | 1.700 | | TS "Allison" 6/24 |
| 1994 | 48 | M | 1.420 | 1.046 | 1.407 | TS "Alberto" 7/03, TS "Beryl" 8/16 | TS "Alberto" 7/03, TS "Beryl" 8/16 |
| 1961 | 48 | L | 1.505 | 1.156 | 1.496 | | H "Carla" 9/9–15 |
| 1999 | 40 | M | 0.932 | 1.042 | 0.950 | | No TC rainfall |
| 2003 | 41 | L | 2.019 | 1.130 | 1.966 | | TS "Bill" 6/27–7/03, TD "Seven" 7/25 |
| 1971 | 36 | L | 1.464 | 1.227 | 1.455 | | TD "2a" 7/5, TD "Eleven" 8/29, H "Fern" 9/1 |
| 1965 | 32 | L | 1.128 | 1.11 | 1.139 | TS "Unnamed" 6/11 | TS "Unnamed" 6/11 |
| 1946 | 32 | L | 1.503 | 1.112 | 1.481 | | No TC rainfall |
| 1983 | 31 | L | 1.142 | 1.013 | 1.140 | | No TC rainfall |
| 1987 | 31 | M | 0.643 | 1.040 | 0.676 | | TS "Unnamed" 8/09 |
| 2013 | 31 | L | 0.723 | 0.864 | 0.736 | | TS "Andrea" 6/05 |
| 2005 | 30 | L | 1.425 | 1.293 | 1.399 | | TS "Arlene" 6/10, H "Cindy" 7/03, H "Dennis" 7/08, H "Katrina" 8/24 |
| 1958 | 29 | L | 0.898 | 1.013 | 0.918 | | No TC rainfall |
| 1975 | 23 | L | 1.295 | 1.123 | 1.300 | H "Eloise" 9/13 | TD "4" 7/27, H "Eloise" 9/13 |
| 1991 | 22 | L | 1.841 | 1.087 | 1.808 | | No TC rainfall |
| 2004 | 22 | M | 0.893 | 1.013 | 0.911 | H "Frances" 9/07 | H "Ivan" 9/26 |
| 1955 | 22 | M | 0.306 | 0.735 | 0.329 | | TD "Brenda" 8/1 |
| 1953 | 21 | L | 1.003 | 0.974 | 1.016 | TS "Alice" 6/07, H "Florence" 9/26 | TS "Alice" 6/07, H "Florence" 9/26 |
| 1964 | 20 | M | 1.084 | 0.959 | 1.085 | H "Dora" 9/11 | TS "Unnamed" 6/4, H "Dora" 9/11 |
| 1972 | 20 | L | 0.763 | 1.064 | 0.781 | H "Agnes" 6/20 | H "Agnes" 6/20 |
| | | | 0.958 | 0.945 | 0.960 | | |
| | | | 0.408 | 0.203 | 0.396 | | |

[1] Proportion of trees indicating a false ring (FR%), location of FR within the annual ring (type), earlywood width (EW), latewood width (LW), total ring width (TRW), tropical cyclones passing within 50 nautical miles of the center of the watershed (tropical cyclone (TC) 50 nautical miles (NM)) and TCs causing heavy rainfall in the basin (TC rainfall data). Date format is month/day. Mean and standard deviation (respectively) of tree ring width parameters for the 67 years analyzed are given in the last two rows.

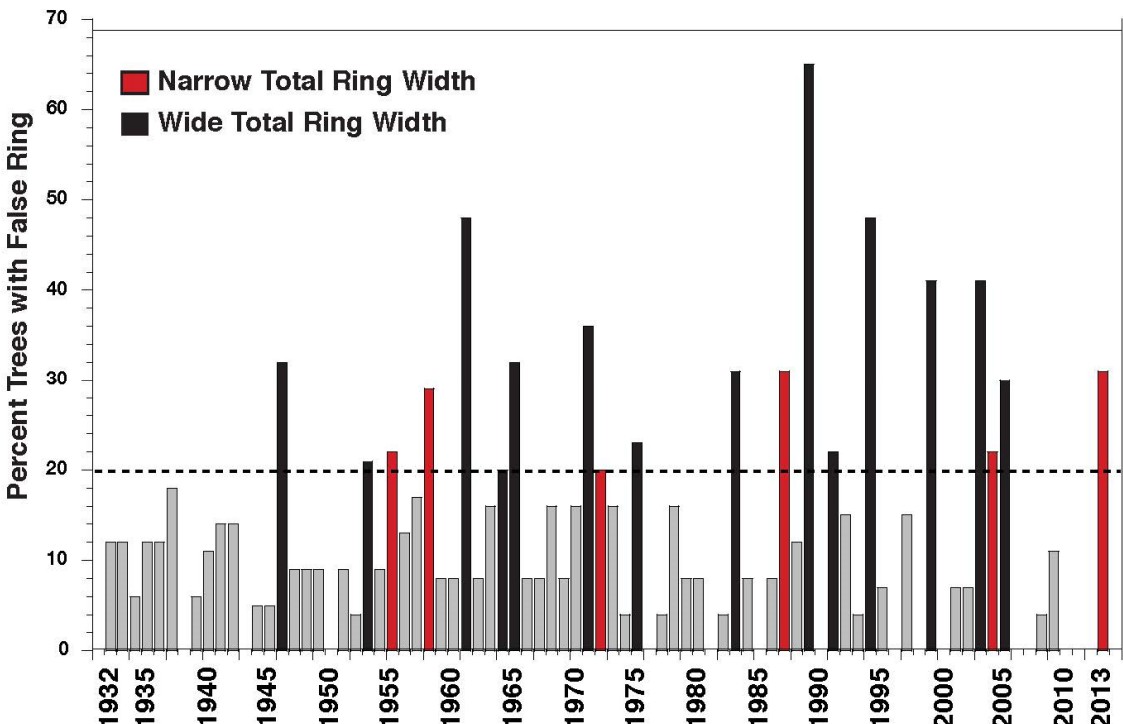

**Figure 3.** False ring response index chronology. Vertical bars indicate the proportion of trees exhibiting a FR in a given year. Black (wider than average rings) and red (more narrow than average rings) bars indicate years in which at least 20% of trees display a false ring (the top quartile of years). Grey bars indicate years (wide and narrow) in which less than 20% of sampled trees show a false ring.

*3.3. Hydroclimatic Significance of False Rings*

The results of the SEA indicate that HFR years were associated with higher than average streamflow in the late spring and summer months, particularly July and August (Figure 4). This was supported by the fact that during the 20 HFR years, discharge in June, July and August was proportionally well above the mean. For example, July discharge averaged over the 20 HFR years was nearly double the average of all years (160 $m^3$/s vs. 284 $m^3$/s; Figure 5), and only two (1955 and 1972) of the 20 FR years had lower than average July discharge. SEA did not indicate any difference in March or April flows, nor were mean discharge conditions lower than normal in any months.

Examination of the daily discharge data (Figure 5) indicates that streamflow in both HFR and all other years typically began to decline precipitously beginning about mid-April (from roughly 300 $m^3$/s) though mid-June when they became relatively stable (140 $m^3$/s). However, in the 20 HFR years dramatic increases in discharge had occurred in late June through August. This pattern was consistent with the association between HFR and high monthly flows in July demonstrated by the SEA. Additionally, one can observe in Figure 5 that while daily discharge in HFR years corresponded with wider than average ring width relative to the non-HFR years, the decrease in daily flow followed by high July/August flows was still apparent. The location of the FR within the annual ring also appeared to be associated with the timing of discharge increases. Figure 6 indicates that summer discharge tended to peak in July during the HFR years placed in the "middle" category and in August for those placed in the "late" category (Table 2).

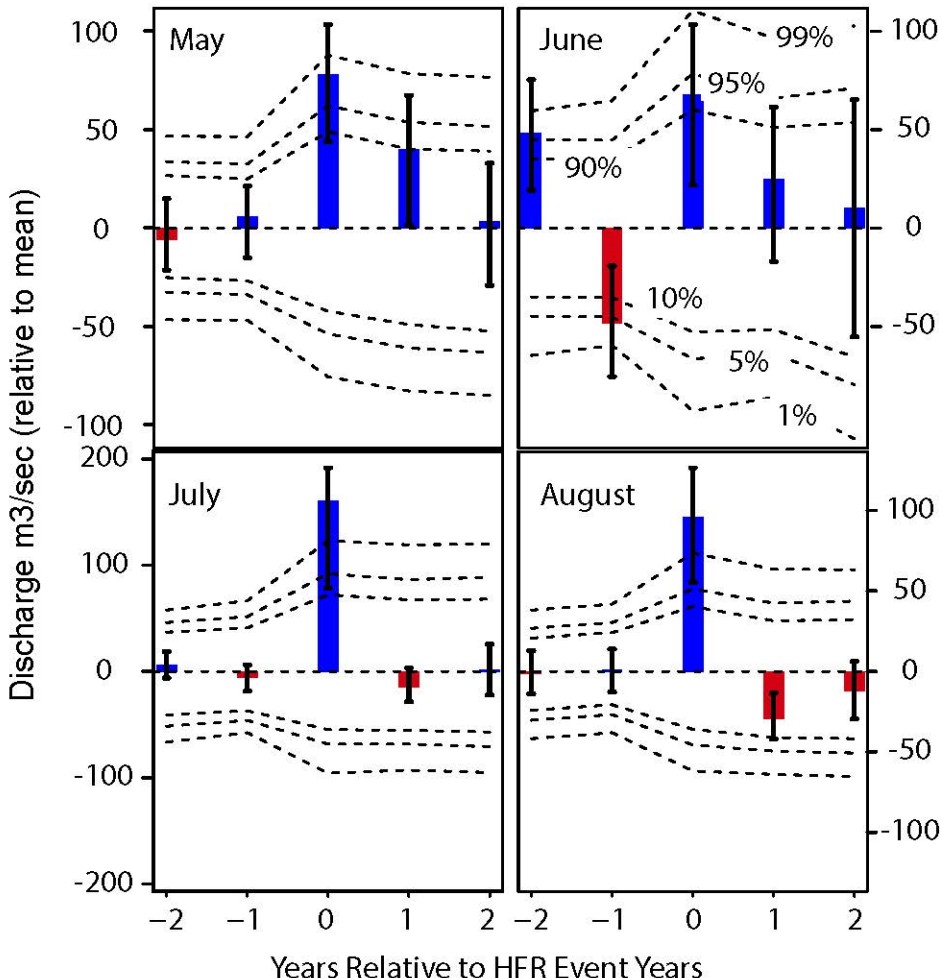

**Figure 4.** Superposed epoch analysis of the H20 "high" FR (HFR) years vs. May–August monthly streamflow. The dashed lines indicate the statistically significant thresholds (1%, 5%, 10%, 90%, 95% and 99%), while the uncertainty intervals are the 5th and 95th percentiles of monthly discharge.

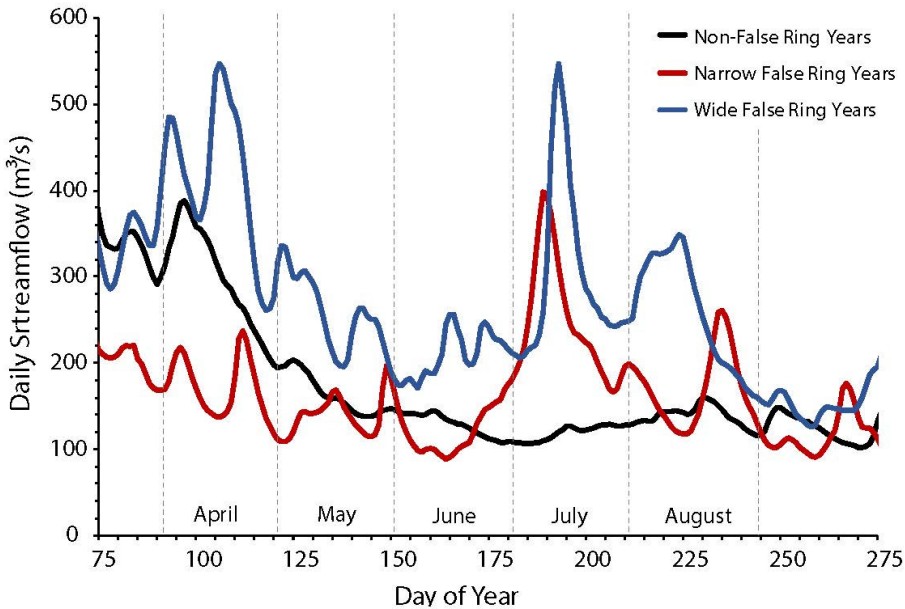

**Figure 5.** Average daily streamflow (day of year) for non-HFR years (black line), narrow HFR years (red line) and wide HFR years (blue line).

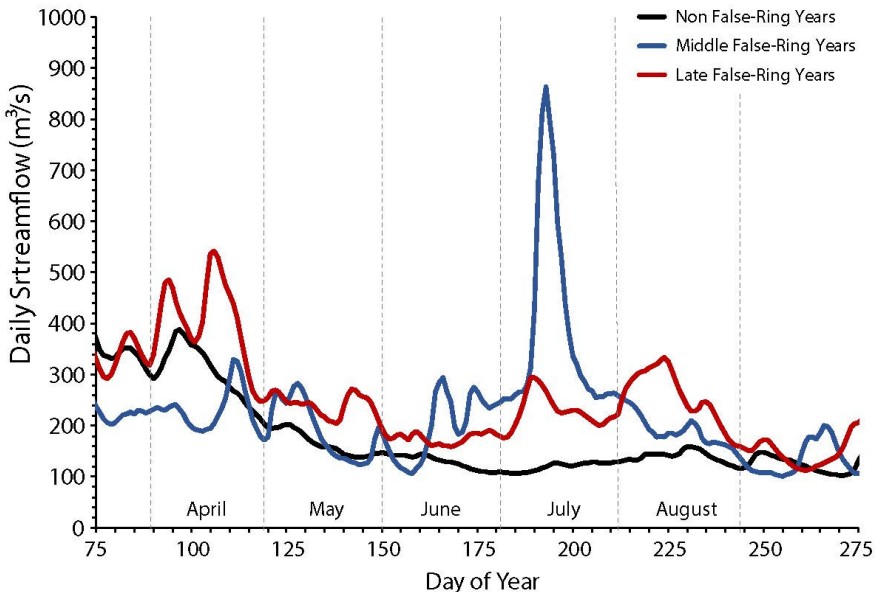

**Figure 6.** Average daily streamflow (day of year) for non-HFR years (black line), HFR years classified as "middle" (blue line) and "late" (red line). Note the middle-years' July peak is especially pronounced due to the effect of Tropical Cyclone "Alberto" in 1994.

Although no year was classified as "early", some years had multiple FR in the "early" position. For example, 1989, which had the highest proportion of trees (65%) with a FR for the entire record, had 30% of FR in the "early" position. Additionally, in 1989 streamflow peaked in June but remained high through early August (Figure 7).

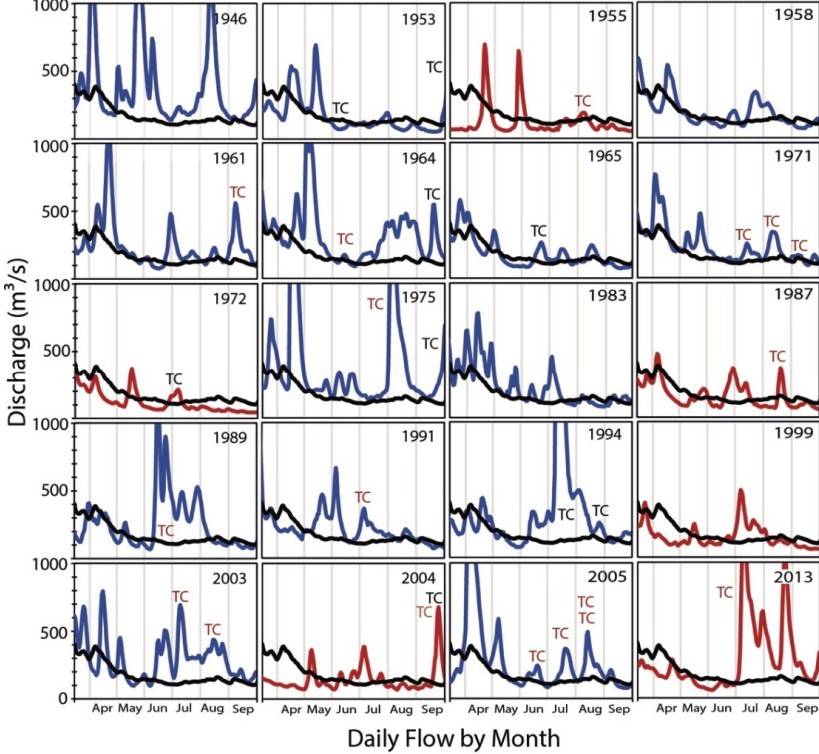

**Figure 7.** Daily streamflow values by month for individual HFR years (blue line = wide rings, red line = narrow rings) and non-HFR years (black line). Low spring discharge followed by above normal summer flows is clear in narrow ring years. Additionally, shown are TC events (black indicated in HURDAT data, red from TCR data).

### 3.4. Estimating Streamflow with Ring Width and FR

GLM of TRW versus monthly discharge indicates that TRW explains as much as 43% of the variance (adj. $R^2$) in summer (JJA) streamflow (Table 3). The inclusion of the binary $FR_b$ variable increases the variance explained (adj. $R^2$) for this period to 49%, providing a modest increase of 14% for the overall improvement in the variance explained with the $FR_b$ inclusion, rather than TRW alone (Table 3, Figure 8b).

**Table 3.** Variance explained (adjusted $R^2$) of the TRW and TRW + $FR_b$ generalized linear models of log-transformed streamflow (Q) in June, July, August and total June, July and August (JJA) discharge.

|  | Variance Explained, $R^2$ | | | |
|---|---|---|---|---|
|  | June Q | July Q | August Q | Total JJA Q |
| TRW | 61 | 30 | 19 | 43 |
| TRW + $FR_b$ | 61 | 43 | 22 | 49 |
| Improvement (%) | 0 | 43 | 16 | 14 |

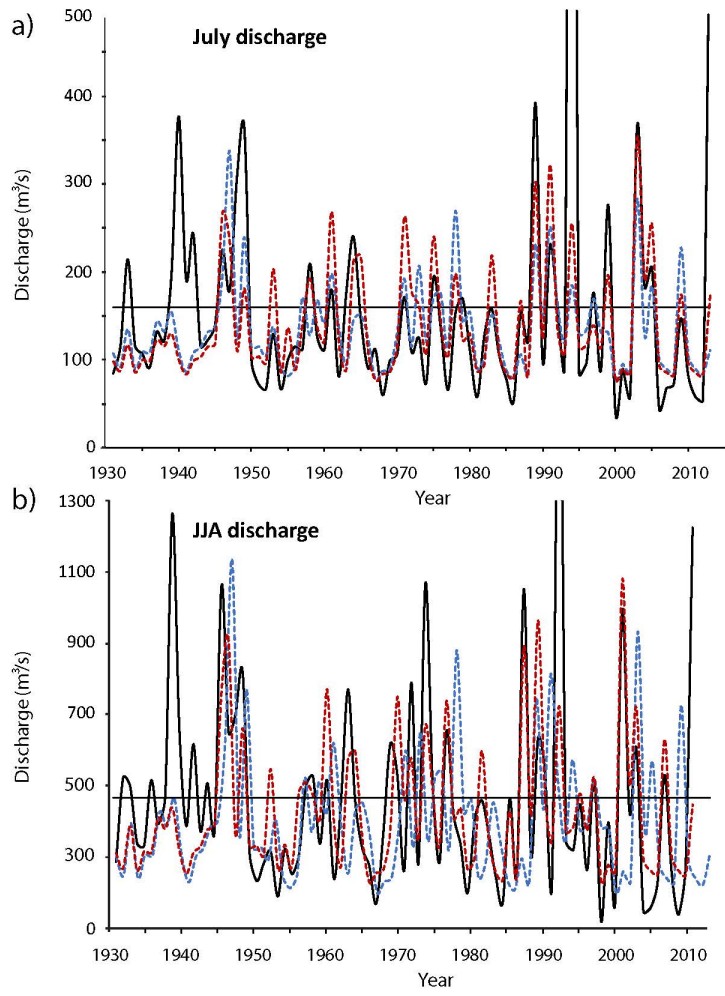

**Figure 8.** Time series of observed (**thick black line**), total ring width-modeled (**blue dashed line**), total ring width + $FR_b$-modeled (**red dashed line**) and observed mean discharge (**thin black line**) for (**a**) July and (**b**) total JJA discharge. Note, all discharge values were log-transformed to account for outliers in the data prior to modeling, with modeled values inverse-transformed prior to plotting. Including the $FR_b$ parameter improves discharge estimates by (**a**) 43% and (**b**) 14%, respectively. Most of the model improvement can be attributed to higher estimated discharge in years with narrow rings and FR. Note: extreme values in 1994 and 2013 are truncated. Figure A2 shows the full range of values.

The greatest model improvement using the $FR_b$ occurred in July, where there was a 43% improvement in the estimation of streamflow using both TRW and $FR_b$ rather than TRW alone (Table 3, Figure 8a), with total JJA modeled streamflow improving by 14% with the addition of the $FR_b$ variable (Table 3, Figure 8b).

### 3.5. False Rings and Tropical Cyclone Activity

Based on the track data, there did not seem to be any clear association with TC events that occurred farther than 50 NM and only moderate association with those that track within 50 NM. Eighteen TCs (wind speed ≥17 m/s) tracked within 50 NM of the watershed center during 15 years from 1931 to 2014 (Table 2, Figure 7). Nine of these 18 TCs occurred during seven HFR years. Specifically, of the three June events (1953, 1965 and 1972), all three were HFR years, and one (TS Alberto, 1994) of the two July events (1970 and 1994) coincided with an HFR year. Of the five August TC events, one (1994) was an HFR year, while two (1936 and 1937) were FR years. Four of eight September TC years (1953, 1964, 1975 and 2004) coincided with HFR years.

Examination of the TC activity based on TC rainfall [46,47] during the 20 HFR years indicates that FR were at least associated with TC rainfall in most (16/20) instances (Table 2, Figure 7). Over the 20 HFR years there were 25 TCs (over 16 years) that produced heavy rainfall over at least portions of the basin. There were seven cases in which the TC rainfall data indicated that heavy rainfall occurred over the basin but those TCs did not track within 200 NM (1955, 1961, 1971, 1989, 1991, 2004 and 2005). There were also eight cases in which a TC did track within 200 NM of the basin but did not cause relevant rainfall as indicated in the TC rainfall data. One of these TC events (Hurricane Frances, 2004) were within 50 NM, yet it did not produce a significant increase in overall discharge (Table 2; Figure 7).

## 4. Discussion

The results presented indicate that the formation of FR at our study site appeared to be a common occurrence. During the instrumental period (1931–2014) there were 67 years that showed some sign of a FR and 20 years in which 20% or more of the sampled trees produced a FR (indicating a HFR year). It also seemed clear that FR are a coherent response by the sampled trees to increases in streamflow in the latter portion of the growing season, which on average extended from June to August. The general pattern for the development of a FR began with a normal seasonal decline in discharge (though not necessarily to below average conditions) in the spring (March–May) season followed by sharp increases in flow in summer (particularly July). While the association between these FR events and high summer monthly flow values was generally apparent, examination of daily flow data was necessary in some cases to observe the relevant changes in flow. For example, in the two cases where the monthly data did not show a wet July (1955 and 1972), daily flows indicate that there were extended multiday periods when daily streamflow was below average that were followed by sharp peaks of increased streamflow (Figure 7). These daily variations were muted by low/high flows in the monthly data but were likely driving the formation of FR in these years.

The fact that most (14) HFR years co-occurred with wider than average TRW and greater than average streamflow throughout both the spring and summer months (AMJJ; Figure 7) further supports the notion that a stressor event is not necessary for the formation of a FR. This is perhaps not surprising given the known response of the species (at least in many instances) to increased streamflow/inundation [49]. Additionally, the relationship between TRW and FR occurrence provides useful information about streamflow variability. For example, in the six HFR years (1955, 1958, 1987, 1999, 2004 and 2013; Figure 7) that co-occur with narrower-than-average TRW it was clear that drier than average spring (March–May) conditions were followed by increased streamflow and reactivation of cambial growth during summer (predominately July; Figure 7). This allows the use of FR information in conjunction with RW to better estimate seasonal variations in streamflow.

Another issue related to the comparison of FR and RW is the determination of the EW/LW boundary. In our analysis most HFR years (16) had wider than normal LW, but because only 14 of

20 years had wider than normal TRW, the methodology of defining the EW/LW boundary [33,34] may be affecting this outcome. We did not compare alternative methods for the measurement of EW/LW [39], but there may be some value in doing so [11]. In addition to affecting EW/LW measurement, it also appeared that the location of FR within the annual ring contained potentially important seasonal climate information. As shown in Figure 6, the location of FR within the annual ring appeared to be associated with peaks in summer streamflow. Although we did not analyze this relationship statistically in this study, it appeared that there might be value in collecting this information while developing FR chronologies and perhaps using it to estimate subseasonal timing of summer streamflow. Future work in this direction should include better quantification of the placement of the FR within the ring as well as other attributes (e.g., FR width, variations in ring density). This could likely be accomplished using quantitative imaging techniques such as blue light intensity [17].

### 4.1. Use of False Rings for Streamflow Reconstructions

Prior work by Stahle et al. [22] indicates that bald cypress TRW has a significant response to precipitation (positive) and temperature (negative) from about March through July, but that EW and LW (LW$_{adj}$) better capture variability in spring (April–May) and summer (June–July) precipitation respectively. In this study we found that TRW was the best single RW predictor of summer (JJA) streamflow both alone or in combination with the FR$_b$ variable. Model improvements using the FR$_b$ variable rather than TRW alone appeared to be the result of HFR years with relatively narrower rings (Figure 7). These HFR years correlated with lower than normal streamflow during the spring, causing initial narrow ring formation. However, this period of lower than average streamflow in the early spring was subsequently followed by an abrupt increase in streamflow in June, July or August, which resulted in the formation of a FR within the otherwise narrow annual ring (Figure 7). This indicates that for at least a portion of the HFR years (6 out of 20), narrow TRW could coincide with FR occurrence, and therefore the full range of seasonal streamflow would not have been reconstructed when using TRW alone.

For this work, focusing on the inclusion of the binary FR$_b$ variable allowed for the separate determination of the utility of FR as indicators of summer streamflow conditions, as compared to using RW variables alone. The analysis of the FR$_b$ variable in our GLM analysis in the form of a binary (presence or absence of FR in a given year) did provide modest improvement in the variance explained (14%) in modeled streamflow, improving predictive capability during the summer months. Given that June, July and August are important periods for agricultural growth, this improvement through the use of the FR$_b$ for modeling streamflow during the latter portion of the growing season could be useful for better assessments of both past and future vulnerability to extreme hydroclimatic events (e.g., flood and drought).

### 4.2. False Ring Association with Summer Storms/Tropical Cyclones

Our limited analysis attempting to determine the potential connection between FR and TC events suggests that the FR chronology likely reflects TC activity to some degree especially in the early summer period. The association between FR and TC may provide valuable information about the variability of TC activity over long periods, however, more work is clearly needed to clarify the relationship. It also seems clear that the hurricane track data alone are not sufficient for understanding the relationship between TC and FR and that TC rainfall data should also be used since it is the TC rainfall that is increasing discharge and therefore the production of a FR. As an example, the most frequent FR year in the record was 1989 (Table 2) and although high rainfall caused by Tropical Storm Allison in the last week of June (Figure 7) clearly led to high discharge, this TC did not in fact track within 200 NM of the Choctawhatchee basin. In addition, high discharge in early June was the result of a series of frontal systems that caused widespread flooding in Southeast Alabama and Northwest Florida. These frontal and TC events together resulted in the wettest June on record at several stations within the Choctawhatchee watershed [50].

Knapp et al. [51] and Mitchell et al. [18] have reported that TC rainfall produces FR in the latewood of longleaf pine (*Pinus palustris*). In addition to TC rainfall, storm surge and strong winds have also been shown to elicit a strong response in coastal pines (*P. elliottii*) [52,53]. Future research using stable isotopes (e.g., $O^{18}$) may allow a better delineation of FR produced by TC rainfall as compared to those resulting from extratropical systems, and the impact that TC-related storm surge and strong winds may have on FR formation.

## 5. Conclusions

The results we presented support other recent findings that FR occurrence can result from favorable late-growing season conditions—in this case, above average streamflow in summer [16,54], rather than solely from unfavorable conditions earlier in the growth period followed by improved conditions [5,6,8,10]. With that said, a portion of the HFR years (30%) did appear to result from an abrupt shift in streamflow conditions, coinciding with below average streamflow in the early portion of the growing season, followed in the summer months by an abrupt increase in streamflow. The coherent FR response by the sampled trees to the unseasonably increased streamflow has potential to be used to develop long chronologies of FR events and to provide additional subseasonal information alone and in conjunction with chronologies of RW parameters to better estimate past streamflow and other hydroclimate variables. There also appears to be some potential for using FR records to study the variability of TC activity over time.

While there have been an increasing number of studies of FR using various tree species, there have been very few in the Southeastern U.S. Given the great number of species useful for dendrochronology in this region that produce FR, particularly bald cypress, additional studies of this type should be carried out. While not done in this initial study, in future work we intend to utilize methods to provide better quantitative approaches for the identification and measurement of FR variables such as quantitative wood anatomy coupled with image analysis and in-situ field instrumentation of trees and other techniques for understanding the response of trees to short-term (daily) changes in their environment [14,55].

**Author Contributions:** M.D.T. conceptualization, field collection, data analysis, original draft preparation and funding acquisition; E.A.E., data analyses, draft preparation, review and editing; M.D.M., field collections, tree ring chronology development and data analyses; J.C.B. and C.S.T., data analyses, draft preparation, review and editing; G.L.H. and J.T.M., conceptualization and funding acquisition; review and editing; G.A.T., field collections, acquisition of local hydrologic data, review and editing, and funding acquisition. All authors have read and agreed to the published version of the manuscript.

**Funding:** This research was supported by NOAA Mississippi Alabama Sea Grant Consortium, (USM-GRO05007-R/RCE-05), the U.S. Environmental Protection Agency Gulf of Mexico Program (EPA-MX-00D67718-0), the U.S National Science Foundation Paleo Perspectives on Climate Change, P2C2 Program (18059590).

**Acknowledgments:** We thank Adlee Bruner for site access and logistical support, and David W. Stahle for site information. We thank numerous students for assistance in field collections and laboratory analysis in particular Charles Lampman.

**Conflicts of Interest:** The authors declare no conflict of interest

**Appendix A**

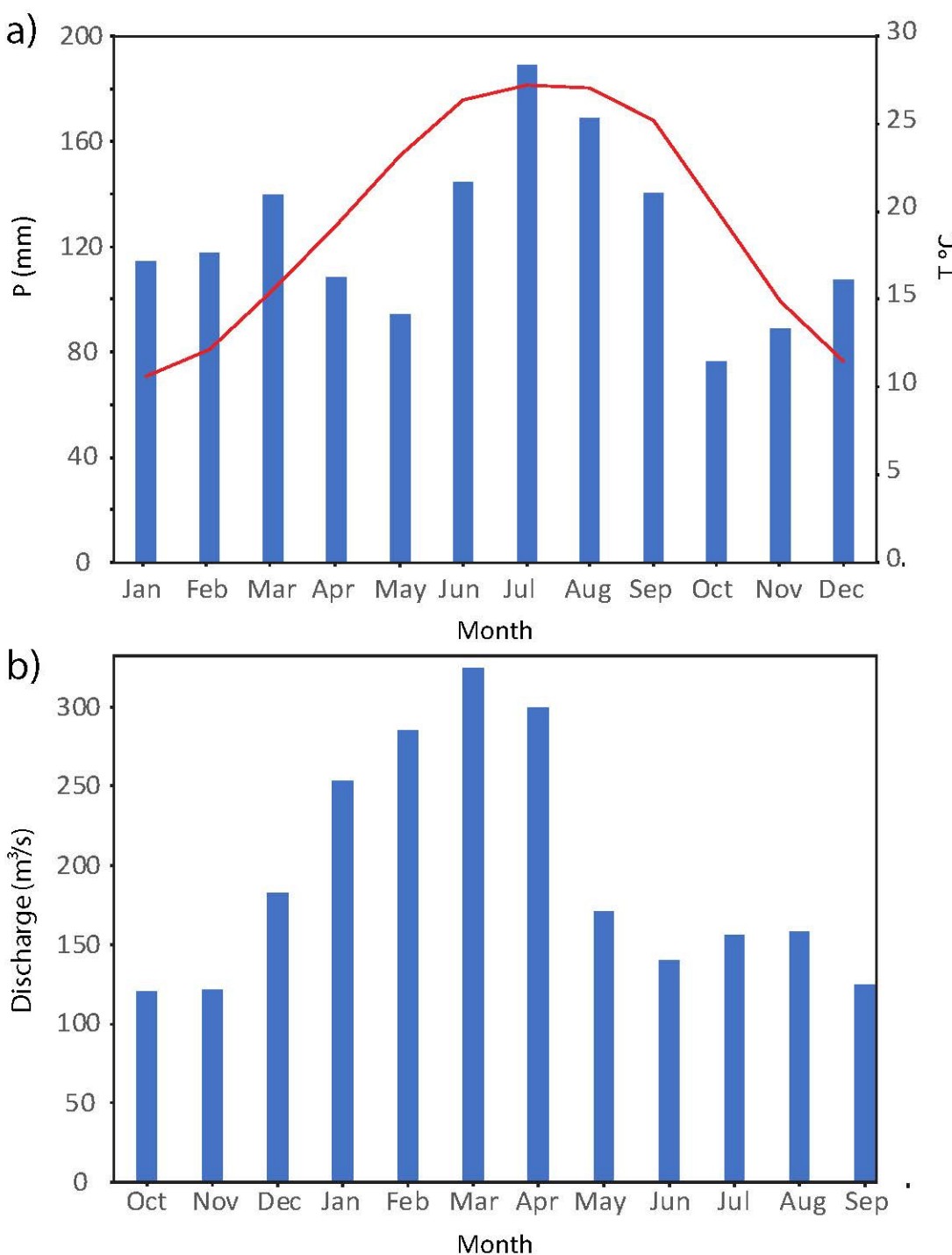

**Figure A1.** (**a**) Climograph showing Florida Division 1 [56] monthly mean precipitation (**blue columns**) in mm and monthly mean temperature (**red line**) in °C and (**b**) hydrograph showing Choctawhatchee River monthly mean streamflow (in cubic meters per second) at the USGS Bruce, Florida gage [27].

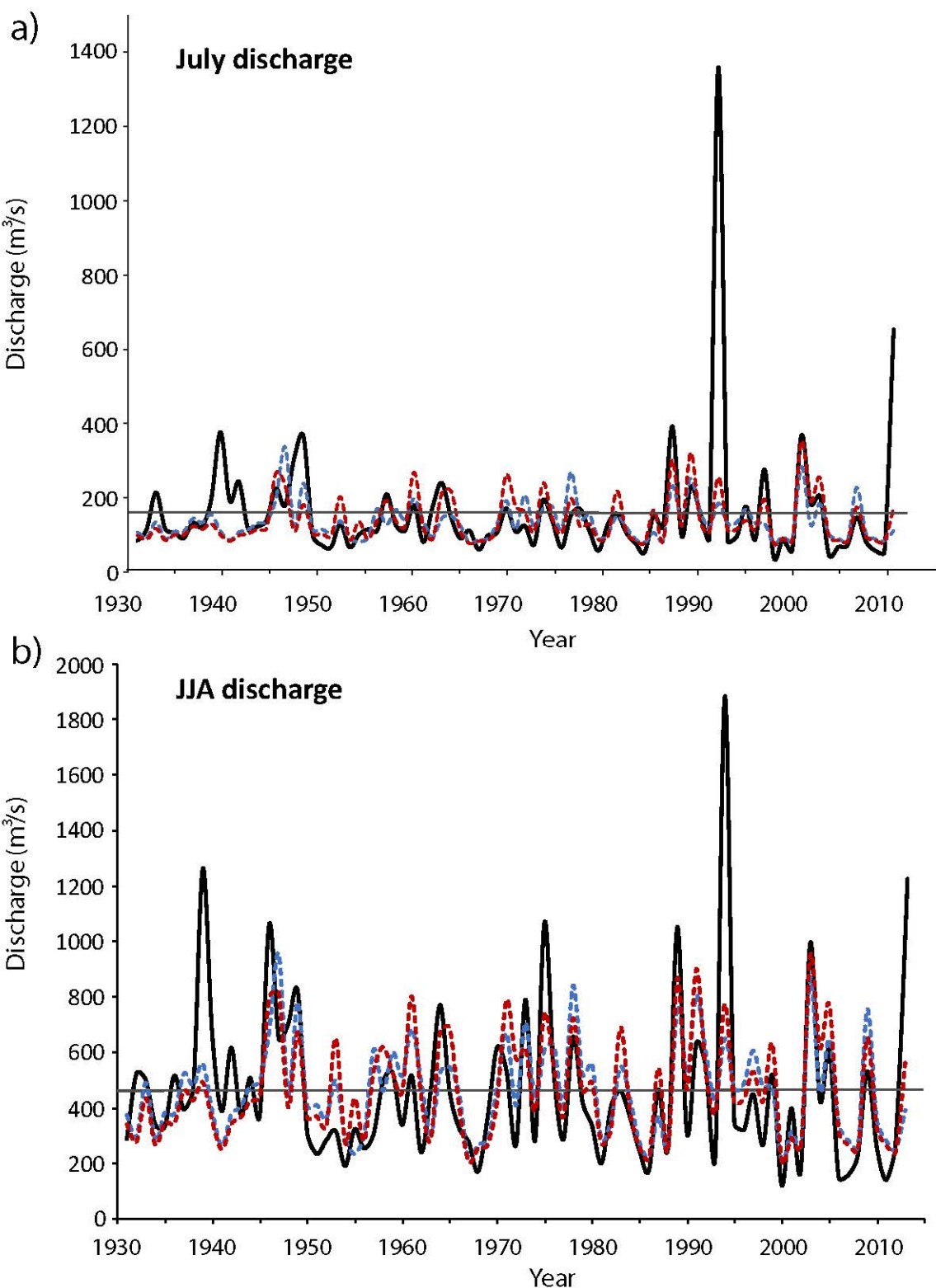

**Figure A2.** Shows the same data as Figure 8 (observed (**thick black line**), total ring width-modeled (**blue dashed line**), total ring width + FR$_b$-modeled (**red dashed line**) and observed mean discharge (**horizontal black line**) for (**a**) July and (**b**) total JJA discharge), but shows the full range of all values. Note the extreme values in 1994 and 2013 caused by rainfall associated with Tropical Storms "Alberto" and "Andrea" respectively.

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
