# Peer review of "Streamflow Variability Indicated by False Rings in Bald Cypress (Taxodium distichum (L.) Rich.)"

_forests, doi:10.3390/f11101100_

Round 1

Reviewer 1 Report

line 50, 51 and further - various references to literature, once there is a space, and sometimes no, correct the edition
line 88 - unnecessary reference to Fig. 1 - does not follow from the drawing that there are two test plots
lines 95-120 - 14 times you write "we", too much, please use the impersonal or passive form
line 107, 108 - formula not readable - put a space under/in the root ??
line 132 - which means LWadj - explain
line 151 formula 2 - the designation of the formula (2) is repeated, delete, in the formula as in the text 0,1 and 2 give as subscripts
line 156 before ..., and give spaces
table 2 - in which units are EW, LW and TRW - number or % or other - complete
fig 3. - what do the gray bars and their absence mean (e.g. 1938), add an explanation in the caption, an unnecessary title at the top of the chart, there is a caption and that's enough
fig 4. - remove the second sentence from the caption - this is described in the text, and add an explanation of what the dashed lines mean (1,5,10,90,95 and 99%). Unnecessary caption at the top of the drawing, delete, also add what whiskers means (STD ??)
fig 5. - removing the figure caption at the top, repeating and removing the second sentence - this should be in the text, not here.
line 237 - not only August, but April too?? - there is also an increase in flows
fog. 6 - delete the caption and the second sentence above - as in fig. 5.
table 3 - what does Q mean for the month ??, explain
fig 8 - a) change the scale (break it) so as to show the discharge values ​​in the 90s and 2010, in the signature ringwidth - insert a 2x space (ring width)
line 298 - removing the dot at the end of the chapter name
line 314 insert a space before 2004
line 338 - use full name or as in line 323 (March-May)
line 362 you write the US once, once the U.S. ?? (vs. line 399)
line 381 - use italics for a Latin name
line 382 - add the Latin name of the species (coastal pine)
line 389 and 390 incorrect parentheses when referring to a bibliography
line 405 to add [14,55], or the same everywhere with or without spaces

Reviewer 2 Report

The manuscript by Therrell et al. deals with the occurrence of false rings (intra-annual density fluctuations, IADFs) in tree rings of bald cypress. The occurrence and position of these IADFs in tree rings is related to streamflow variability.

The subject is worth of investigation and the data shown are interesting, but the paper needs revision throughout the text as reported following.

Title

  1. It should be changed to reflect the contents of the study. The terms “false ring formation” must be avoided because xylogenesis is not analysed.

Abstract

  1. The last sentence should be rephrased.

Introduction

  1. The introduction begins with a sentence that describes the false rings. However, this sentence describes only one type of IADFs (latewood-like cells in earlywood), while these fluctuations have been classified into different types in papers that are also referred to by the authors later on. Please, rephrase the first part of the Introduction to consider all types of IADFs (e.g. earlywood-like cells in latewood).
  2. There is lack of context: why this study should attract the interest of the readers of Forests?
  3. Lines 47-48. This statement is not true. There are many examples in literature in which environmental signals hidden in IADFs are unravelled. This is surely true for Mediterranean species.
  4. Lines 55-56: not clear, rephrase.
  5. Lines 66-72. Please avoid reporting conclusions in the Introduction

Methods

  1. Figure 2. It is not clear whether the late fluctuation in c and d is considered as latewood-like cells in earlywood or earlywood-like cells in latewood. Please consider this (also taking into account comment n. 3) because the functional meaning of the two types of fluctuations can be different. At least, comment on this.
  2. Lines 88-93. A climatic diagram would be useful.
  3. Lines 115-117. Multiple false rings should be considered separately. What is the rationale behind the decision to categorize such false rings on the basis of their first appearance?

Results

  1. Results should be carefully checked because they should just report the findings without comments and thus references (examples are lines 258-259, 266-269, 284-290.
  2. Table 2. Statistics are not shown (at least mean values and standard errors).
  3. Figure 3. What is the thresholding to classify a ring between the two classes “narrow” and “wide”? This figure should be better commented in the text.
  4. Figures 5-6-7 are commented in a different order in the text and explanation is not exhaustive.

Discussion

  1. The discussion is poor and is more like a summary of findings. The first 14 lines are an example.
  2. Lines 342-345. This statement raises some concerns on the identification of the boundary between earlywood and latewood and thus on the reliability of part of the results reported in this study. References 33-34 are not appropriate. There is a clear definition for latewood that is commonly applied and it is not related to the onset of lignification: indeed cell walls of earlywood conduits are lignified as well. The Mork’s low must be applied to classify latewood, and thus to properly detect the ratio EW/LW: all tracheids whose common double cell wall is equal to greater than the cell lumen are considered latewood (parameters always measured in the radial direction) (Mork, E. 1928. Die Qualität des Fichtenholzes unter besonderer Rücksichtnahme auf Schleifund Papierholz. Der Papier-Fabrikant 26: 741–747; this paper is referred to in many papers dealing with the classification of EW and LW).
  3. Lines 351-354. This sentence here is not well integrated.
  4. In general, the authors make the effort to classify the different types of IADFs based on their position but the relations with environmental factors and their functional meaning are poorly discussed.

Reviewer 3 Report

General Comments

This article takes advantage of a little used characteristic of false rings in bald cypress to reconstruct extreme and short-term events like flooding from tropical cyclones. It is one of a few studies that have found these event markers of shorter term events buried in the more continuous and long term data that we normally collect from tree rings, making this an interesting article with broad importance. These findings are likely to lead to more and longer-term studies that can reconstruct tropical cyclones prior to good-quality historical record keeping.

I don't like the development of acronyms just for a single paper. I would suggest that you write out false rings and high false ring chronology. This is a style decision for the journal, but the paper would be more accessible and easier to read if you wrote out false rings, high false ring chronologies, and tropical cyclones.

Line 50: Large and small scale are oppositely defined in geography and ecology and your research fits well into these two fields.  Use broad and fine scale instead.

Lines 68-71: This sounds like you main conclusion at the end of your introduction. I suggest that you end your introduction with your hypotheses rather than your conclusions.

Line 69: Utilize has a specific chemical meaning of using something up in a reaction. You should just use the word "use".

Section 2.1: You need more than once sentence to form a paragraph. Combine this with the following paragraph and move the graphic to the next page.

Figure 1: What are the lines on the map? I assume one is a county line and the other may be the coast? Proximity to the coast can be an important factor in tree growth, so clarify this please. Also, I would zoom in so that you can see the separated BRI and JLK sites.

Line 85: You list two sites, but only have one site mentioned and marked on the map. Are the sites so close together that they don't separate from each other on the scale of your map

Line 88: You mention BRI and JLK and reference figure 1, but don't show these sites (as separate symbols on the map. Can you have a zoomed in inset to show the relationship of the two sites to each other? Otherwise, what is the purpose of the separated sites if they are too close together to differentiate on your map?

Line 88-93: Separate this last bit into a different paragraph as you have changed the subject from your site description to the climatic description.

Lines 105-106: I understand that you mainly used the event chronology of false rings but did you standardization you EW, LW, and TRW chronologies?  If so, what standardization did you use?

Figure 5 and 6: I am not sure what the difference between figure 5 and 6 is. According the the legend, one is average daily streamflow while the other is mean observed daily discharge. Both have figure title of Average Daily Streamflow. Please clarify.

Specific Comments

Line 51: Please cite the two studies that you are referring to at this location.

Line 96: 5.15mm Swedish Increment borer.

Line 389 and 390: Need a close square bracket "]" instead of an end parentheses ")".  Although the e.g. statement still needs the end parentheses (probably the first one). 

Round 2

Reviewer 2 Report

Following the comments to Authors responses:

Comment 4. There is lack of context: why this study should attract the interest of the readers of Forests?

Response 4. We respectfully disagree with the Reviewer. This manuscript relates to several subject areas of Forests (e.g., forest ecology, ecophysiology and biology, and climate change impacts). Furthermore, tree-ring focused articles are routinely published in this journal. This manuscript describes the most thorough and complete study of an important climatic response of one of the most important species in the region both in terms of ecological, and economic value, and certainly the most valuable in terms of paleoclimate research. I am quite certain many readers will find this study extremely valuable.

New Comment 4. This is exactly the point! I fully agree that the contents of this paper are relevant for Forests and are of high impact but this is not enough underlined in the Introduction. The things that Authors reported in the response to my comment should be reported in a couple of sentences in the Introduction.

Comment 6. Lines 55-56: not clear, rephrase.

Response 6. I am not clear on what needs rephrasing. Perhaps the reviewer can be more specific.

New Comment 6. The meaning of this sentence is not clear. Could you please try to rewrite it?

Comment 7. Lines 66-72. Please avoid reporting conclusions in the Introduction

Response 7. We respectfully disagree with the reviewer. It’s common to report a short summary of findings in the Introduction. For example, see Meko, D.M.; Touchan, R.; Kherchouche, D.; Slimani, S. Direct Versus Indirect Tree Ring Reconstruction of Annual Discharge of Chemora River, Algeria. Forests 2020, 11, 986.

New Comment 7. We could report many examples of papers in which conclusion “are” or “are not” reported at the end of Introduction. No specific rules are reported in the Instructions of Forests, but I still believe it should be avoided. Batter save some lines to add a couple of sentences on the context and relevance as reported in comment 6.

Comment 11. Results should be carefully checked because they should just report the findings without comments and thus references (examples are lines 258-259, 266-269, 284-290).

Response 11. I am not sure I understand the issue here. In each of these cases we are providing context for our findings in an attempt to better understand the relationship between false rings and climate.

New Comment 11. Indeed “context of findings in an attempt to better understand the the relationship between false rings and climate” must be part of the discussion. References are generally not reported in the Results section. Please limit results section to show the data. Data interpretation and comments are part of the discussion.

Comment 15. The discussion is poor and is more like a summary of findings. The first 14 lines are an example.

Response 15. I’m not sure what sort of reply is requested here. I don’t think it is unusual to provide some sort of summary discussion in this section.

New Comment 15. In this manuscript, there is a short summary of main findings in the end of Introduction and at the beginning of Discussion: maybe this is too much. I am aware that a short summary at the beginning of the Discussion can be helpful but this is 20% of the whole discussion! Please, consider this comment together with comment n. 11: remove comments and interpretation from results and use those sentences to enrich the discussion.

Comment 16. Lines 342-345. This statement raises some concerns on the identification of the boundary between earlywood and latewood and thus on the reliability of part of the results reported in this study. References 33-34 are not appropriate. There is a clear definition for latewood that is commonly applied and it is not related to the onset of lignification: indeed cell walls of earlywood conduits are lignified as well. The Mork’s low must be applied to classify latewood, and thus to properly detect the ratio EW/LW: all tracheids whose common double cell wall is equal to greater than the cell lumen are considered latewood (parameters always measured in the radial direction) (Mork, E. 1928. Die Qualität des Fichtenholzes unter besonderer Rücksichtnahme auf Schleifund Papierholz. Der Papier-Fabrikant 26: 741–747; this paper is referred to in many papers dealing with the classification of EW and LW).

Response 16. I understand the reviewer’s statement and am aware of the methods used to classify latewood in the tree-ring literature more focused on wood anatomy. However in this case we are using methods established and frequently used by the dendroclimatology community. The relevant references are certainly appropriate in this case and many in our discipline will be quite familiar with them. However, I also realize that the phrase “first onset of lignification” is not the best term to use here and so have removed that.

New Comment n. 16. The community of scientists working on tree rings is making the effort to move from descriptive to quantitative data otherwise misleading results can be achieved. This is the reason why the dendroclimatology community is working closer and closer to quantitative wood anatomists. Removing at least the reference to onset of lignification is useful as it could be useful to better state how EW and LW were visually classified.

Comment 17. Lines 351-354. This sentence here is not well integrated.

Response 17. I’m afraid this is not specific enough for me to reply to.

New Comment 17. This is a very general statement and at this point of Discussion is not needed.
